

# Spatial variability and sampling density of chemical attributes in archaeological black earths under pasture in southern Amazonas, Brazil

Alan F. L. de Lima[1], Milton C. C. Campos[1], José M. da Cunha[1], Laércio S. Silva[2], Flávio P. de Oliveira[3], Bruno C. Mantovanelli[4], Elilson G. de Brito Filho[1], Romário P. Gomes[2]

[1]Universidade Federal do Amazonas (UFAM), Humaitá, Amazonas, 69800-000, Brasil
[2]Universidade Estadual Paulista (UNESP), Jaboticabal, São Paulo, 14870-180, Brasil
[3]Universidade Federal da Paraíba (UFPB), Areia, Paraíba, 58397-000, Brasil
[4]Universidadae Federal de Santa Maria (UFSM), Santa Maria, Rio Grande do Sul, 97110-566, Brasil

*Correspondence to*: Milton C. C. Campos (mcesarsolos@gmail.com)

**Abstract.** Spatial mapping of soil chemical attributes is essential for sampling efficiency and agricultural planning management, ensuring a regional development and sustainability of the unique characteristics of archaeological black earths (ABEs). Thus, this study was developed aiming at assessing the spatial variability and sampling density of chemical attributes in soils of ABEs under pasture in southern Amazonas, Brazil. A sampling grid of $56 \times 80$ m with regular spacings of 8 m was installed in the experimental area and samples were taken from the crossing points at depths of 0.0–0.05, 0.05–0.10, and 0.10–0.20 m, totaling 264 georeferenced points. The chemical attributes pH in water, organic carbon, Ca, Mg, K, P, Al, and potential acidity were determined in these samples, while CEC, SB, V, t, T, and m were calculated. The attributes present a spatial dependence varying from strong to moderate, being Al3+ the only chemical attribute that does not present a spatial dependence structure in the assessed depths. Scaled semivariograms satisfactorily reproduce the spatial behavior of attributes in the same pattern of individual semivariograms, allowing their use to estimate the variability of soil attributes. Sampling density is higher at a depth of 0.0–0.05 m, requiring 2 and 1 point ha−1 at depths of 0.05–0.10 and 0.10–0.20 m, respectively, to represent the spatial pattern of chemical attributes.

## 1 Introduction

The exploitation of regional potential is one of the alternatives to ensure a sustainable agricultural success (Oliveira et al., 2014, Gomes et al., 2017, Brito et al., 2018). In this scenario are inserted the archaeological black earths (ABEs) or Indian black earths present in the Amazonian landscape. These earths are found on the banks of the rivers Purus, Madeira, Juruá, Solimões, and Amazonas (Kern et al., 2003) and stand out for being soil units with several remarkable characteristics such as the dark coloration and presence of ceramic and/or lithic fragments incorporated into the matrix of the surface soil horizons (Petersen et al., 2001). From the chemical aspect, these soils present a high fertility when compared to the vast majority of



soils typical of the Amazon region, with high total contents of CaO (1.810 mg kg−1) and P2O5 (4.900 mg kg−1), organic matter, and an intense biological activity (Lehmann et al., 2003, Glaser, 2007).

Chemical quality of ABEs has been an attraction for agricultural exploitation in the Amazon (Silva et al., 2016). However, the lack of detailed studies on these earths is a barrier to the development of precision agriculture (Gomes et al., 2017). Although

some studies have focused on the characterization of ABE attributes (Aquino et al., 2016), few of them have investigated the spatial pattern of its chemical attributes due to the great territorial dimension and diversity of soil, relief, and vegetation of the Amazonas State. In order to take advantage of the regional natural resources without depleting soil chemical quality, it is essential to know the spatial variability of its attributes (Oliveira et al., 2014) since it is a complex interaction of factors and processes of formation, having as additional sources the management of soil, crop, and landscape variations (Campos et al.,

2012; Roger et al., 2014;  Oliveira et al., 2017; Brito et al., 2018). Thus, agricultural success or an adequate and efficient soil management can only be possible when the spatial pattern of its attributes is defined (Gomes et al., 2017).

The variability of soil attributes is often inferred by descriptive statistics (mean, standard deviation, coefficient of variation, etc.). Although these measures of dispersion provide an idea of variation, they do not consider the space, which can often provide misleading information and, consequently, lead to failure of agricultural planning management (Gomes et al., 2018).

In the Amazonas State, some studies such as those of Campos et al. (2012) have been a pioneer in characterizing the spatial variability of soil chemical and physical attributes, as well as have served as a reference for conducting other investigations on this subject. Many of these studies have revealed the potential of ABEs for agricultural purposes (Silva et al., 2016; Gomes et al., 2017; 2018; Brito et al., 2018). However, many of these researchers have warned that an adequate ABE management has altered their natural and desirable chemical properties.

In this scenario, geostatistics is a promising tool for identifying and characterizing soil attributes, considering man-made management in the most varied landscape forms (Gomes et al., 2017). Geostatistics can be used in interpreting and projecting results based on the structure of natural variability, indicating alternatives of use, in addition to allowing a better understanding of attribute distribution and, consequently, its influence on yield (Guan et al. 2017). Moreover, the ideal minimum number of samples representative from the lithological, pedological, and geomorphological diversity of a given region can be determined

from a scaled semivariogram (Teixeira et al., 2017). The combination of these tools maximizes sampling efficiency and reduces costs with manpower.

With the aid of geostatistics, it will be possible to characterize in detail the organic carbon and the chemical attributes of ABEs. From this, to understand the spatial pattern aiming to map all the soils of the Amazonian environments with a reduced number and representative of the pedological, lithological and biodiversity diversity, due to its importance in the world scenario, as for

example in the climatic maintenance of the earth.

Therefore, considering the potential of ABEs for agricultural use, the characterization of the spatial pattern of soil chemical attributes is essential in order to establish a specific soil and crop management, in addition to ensuring the sustainability of natural characteristics of ABEs. Thus, Mapping the spatial pattern of chemical properties of ABE, it will be possible to establish specific areas of management and to support agricultural planning, as well as economical recommendation of fertilizers and





correctives (Roger et al., 2018; Oliveira et al., 2018). For this reason, this study was developed with the purpose of assessing the spatial variability and sampling density of chemical attributes in soils of ABE under pasture in the southern Amazon.

## 2 Material and methods

The study area is located in southern Amazonas State, outskirts of the community of Santo Antônio de Matupi along the BR 230, the Trans-Amazonian Highway, in the region of Manicoré, AM, Brazil, under the geographical coordinates of 7°53′37.71″ S and 61°23′54.64″ W, with an average altitude of 135 m (Figure 1).

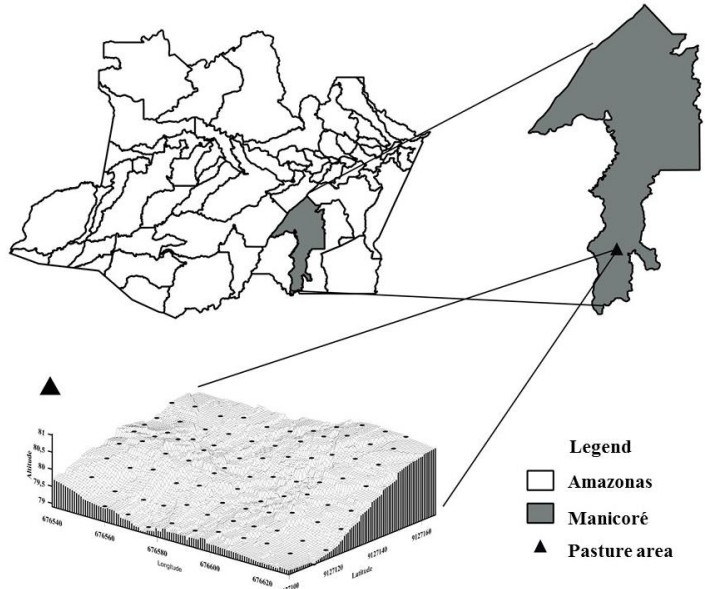

**Figure 1: Study area location, digital elevation model, and collection points of soil samples in archeological black earth areas under pasture in Manicoré, AM, Brazil.**

According to Köppen classification, regional climatic zone belongs to group A (tropical rainy climate) and climate type AM (monsoon precipitation), presenting a dry season of short duration, an average annual precipitation of 2.500 mm, and a rainy season from October to June. The average annual temperature is between 25 and 27 °C and the relative air humidity varies from 85 to 90%. The characteristic vegetation of this region is the dense tropical forest composed of densified and multi-layered trees between 20 and 50 m in height. According to Campos et al. (2012), the predominant regional landscapes are the Natural Fields, Natural Fields/Forests, and Denses Forests.

The study was carried out in an ABE area cultivated with Brachiaria brizantha. This area has a 10-year history of pasture cultivation without any soil tillage to maintain its quality. A sampling grid of 56 × 80 m with regular spacings of 8 m was installed in the area and samples were taken from the crossing points of the grid, totaling 88 sampling points. These points were georeferenced with a Garmin Etrex (South American 69) GPS equipment, with an accuracy < 15 m, 95% typical.





Subsequently, soil samples were collected at depths of 0.0–0.05, 0.05–0.10, and 0.10–0.20 m, totaling 264 soil samples used to determine the chemical attributes.

After collection, samples were air-dried and sieved in order to be analyzed according to the methodology proposed by Embrapa (2011). In this methodology, pH in water is determined by a pH meter with a proportion of soil and distilled water of 1:2.5;

calcium ($Ca^{2+}$), magnesium ($Mg^{2+}$), and exchangeable aluminum ($Al^{3+}$) are extracted by 1 mol $L^{-1}$ KCl solution, with $Ca^{2+}$ and $Mg^{2+}$ contents determined by atomic absorption spectrometry and $Al^{3+}$ by titration; potassium ($K^+$) and phosphorus (P) are extracted by Mehlich-1 and determined by flame photometry and colorimetry, respectively. In addition, potential acidity (H+Al) is determined by extraction with 0.5 mol $L^{-1}$ calcium acetate and subsequent titration. Based on these determinations, we calculated the effective cation exchange capacity (CEC), sum of bases (SB), base saturation (V), and aluminum saturation

(m). Organic carbon (OC) was determined by the Walkley-Black method modified by Yeomans & Bremner (1988).

Soil attributes were analyzed by descriptive statistics, being calculated the mean, median, maximum, minimum, standard deviation, coefficient of variation, and the coefficients of skewness and kurtosis. The hypothesis of normality of the data was tested by the Kolmogorov-Smirnov (KS) test ($p \leq 0.05$) by using the software Minitab 14 (Minitab, 2000).

Geostatistics was used to assess the spatial variability of the studied attributes, as in Vieira et al. (1983). For this, the spatial

dependence was verified by means of the semivariogram graph using the software GS+ version 7. Based on the assumption of stationarity of the intrinsic hypothesis, the semivariogram was estimated by equation 1:

$$\hat{\gamma}(h) = \frac{1}{2N(h)} \sum_{i=1}^{N(h)} \left[ Z(x_i) - Z(x_i + h) \right]^2 \tag{1}$$

where $\gamma(h)$ is the semivariance value for a distance h, N(h) is the number of pairs involved in the semivariance calculation, $Z(x_i)$ is the value of the attribute Z in the position $x_i$, $Z(x_i+h)$ is the value of the attribute Z separated by a distance h from the

position $x_i$.

The experimental semivariograms were chosen based on the number of pairs involved in the semivariance calculation, presence of a clearly defined sill (Burrough & McDonnel, 2000), better coefficient of the cross-validation test, and higher coefficient of determination ($R^2$), in which the values vary from 0 to 1, where those close to 1 characterize the model as more efficient to express the studied phenomenon.

From the adjustment of a mathematical model to the calculated values of $\gamma(h)$, theoretical model coefficients are defined for the semivariogram: nugget effect ($C_0$) is the semivariance value for a distance zero, which represents the component of random variation; structural variance ($C_1$); sill ($C_0+C_1$) is the semivariance value at which the curve stabilizes over a constant value; and range (a) is the distance from the origin to where the sill reaches stable values, expressing the distance beyond which the samples are not correlated (Vieira et al., 1983; Trangmar et al., 1985).

The degree of spatial dependence was analyzed according to the Cambardella et al. (1994) classification, in which semivariograms that have a nugget effect to sill ratio [$C_0/(C_0+C_1)$] lower than or equal to 25% have a strong spatial dependence, between 25 and 75% have a moderate spatial dependence, and higher than 75% have a weak spatial dependence. After





adjustment of the permissible mathematical models, data interpolations were performed by means of kriging in the software Surfer version 13.00.

Scaled semivariograms were also used in order to reduce them to the same scale, making the comparison between the results of different variables easy. The experimental semivariograms were scaled by dividing the semivariance by the sample variance

of each studied variable (Vieira, 1997). With the dimensionless semivariance, the nugget effect expressed directly in sill percentage (total semivariance) the random component of the variance structure. According to Vieira (1997), a proportionality is verified when scaled semivariograms allow the adjustment of a single model for the variable under study.

The scaled experimental semivariograms were adjusted to the spherical model:

$$\begin{cases} \hat{\gamma}(h) = C_0 + C_1\left[\frac{1}{2}\left(\frac{h}{a}\right) - \frac{1}{2}\left(\frac{h}{a}\right)^3\right], \text{ if } 0 < h < a \\ \hat{\gamma}(h) = C_0 + C_1, \text{ if } h \geq a \end{cases} \tag{2}$$

where C0 is the nugget effect, C0+C1 is the sill, h is the separation distance between two observations, and a is the range of the spatial dependence.

Subsequently, the scaled semivariograms served as a basis of information to calculate the minimum number of soil samples and determine the variability of all attributes at different depths.

$$N = A/(a^2/10000) \tag{3}$$

where N is the minimum number of samples required for determining a sampling grid, A is the total area (ha), and a is the semivariogram range (m).

## 3 Results and discussion

The attributes active acidity (pH), calcium ($Ca^{2+}$), magnesium ($Mg^{2+}$), potassium ($K^+$), phosphorus (P), aluminum ($Al^{3+}$), potential acidity (H+Al), and the calculated variables sum of bases (SB), cation exchange capacity (CEC), effective CEC (t),

potential CEC (T), aluminum saturation (m), and base saturation (V) showed values of skewness and kurtosis close to zero, evidencing the symmetry of data at depths of 0.0–0.05, 0.05–0.10, and 0.10–0.20 m (Table 1). Values close to the mean and median indicate that these variables had a normal distribution, a behavior that was verified in the Kolmogorov-Smirnov test. These results are in accordance with those found by Oliveira et al. (2014) in soils of archaeological black earth under forest cultivation.

The average values of chemical attributes (Table 1) indicated a soil with an active acidity of medium toxicity, high potential acidity, low exchangeable aluminum, and very low aluminum saturation, as defined by Embrapa (2013). High pH values from 4.8 - 6.4, according to Falcão et al. (2009), reflect the reduced acidic condition common in ABE. This is associated with processes that change soil acidity, such as organism respiration and decomposition of organic matter, which are very active in ABEs soils. Low concentrations of $Al^{3+}$ show a potential acidity that consists mainly of $H^+$ ions, which is not harmful to plants



(Hernández-Soriano, 2012). This behavior is of great importance from an agronomic point of view since high concentrations of $Al^{3+}$ promote a reduction in P and Ca contents in leaves and roots, thus delaying the availability and absorption of nutrients (Meriño-Gergichevich et al., 2010). A high Al content is not detrimental to plants in soils that have high organic carbon content, as some of the components of organic matter form complexes with Al within soil solution, making it unavailable to plants

(Hernández-Soriano, 2012). Thus, soluble Al is not considered to be a problem for such soils, even if they have a low pH.

High values of pH, OC, P, $Ca^{2+}$, and $Mg^{2+}$ at depths of 0.0–0.05 m, 0.05–0.10 m, and 0.10–0.20 m are related to the anthropic horizon at the soil surface (Aquino et al., 2016). According to these authors, the high contents of $Ca^{2+}$ and P in ABEs are due to the presence of human and animal bones, fish bones, and chelonian shells. For Alquino et al. (2016), the higher P content in ABE is due to the mineralogical composition of the ceramics found in the areas that have high levels of this element. Another

10 hypothesis is that the formation of complexes of cationic ions with high stability OM contributes to ABE chemical richness (Lima et al., 2002; Novotny et al., 2007). According to Oguntunde et al. (2004), the increase in soil pH after the partial burning of pyrogenic coal is attributed to an increase in $Ca^{2+}$ and $Mg^{2+}$ made available in soil by this material. The increase of these cations may have masked the $Al^{3+}$ activity, which is a desirable behavior.

The classification of soil in good and very good classes for P, $Ca^{2+}$, and $Mg^{2+}$ allows considering it with a high fertility level

(Embrapa, 2013) although $K^+$ has presented very low contents. Falcão & Borges (2006) attributed a low productivity of banana and coconut to $K^+$ deficiency when no potassium fertilization was applied in ABEs. Such fact indicates that ABEs do not necessarily have a high availability of all essential nutrients for plants (Lehmann et al., 2003). The excess of $Ca^{2+}$, $P^-$ and $Mg^{2+}$ from ABE soil can establish a nutritional imbalance by competing with $K^+$ for the same cation exchange site. In addition, the retention energy of the $Ca^{2+}$, $Mg^{2+}$ and $K^+$ exchangeable cations to the soil colloids follows a lyotropic series (Tan, 2011),

occupying the fifth element of this series, K becomes less adsorbed to soil colloids. Therefore, in well drained soils, for example, in the Amazon region, the leaching is higher, decreasing the concentration of K in the soil solution.

The abundance in OC content, with an average ranging from 135 to 133 g $dm^{-1}$, is in accordance with the results obtained by Campos et al. (2011), who observed the same behavior in A horizons of Amazonian anthropogenic soils. According to Glaser et al., (2007), the ability of ABE to maintain a high OC content is due to chemical characteristics and resistance of the material

to microbial decomposition. Natural and induced fires can lead to an incomplete combustion of organic material, which leads to the formation of a series of compounds called pyrogenic carbon (Glaser & Birk, 2012). These compounds, given their recalcitrance, represent an important stable carbon reservoir (Glaser et al., 2001). In fact, the custom of ancient indigenous peoples to build large holes to incinerate debris may have provided this OC stability, persisting in the soil until today. This practice is considered the main anthropic process of ABE genesis, as addressed in several studies (Glaser & Birk, 2012; Aquino

et al., 2016).

In general, high CEC, SB and V values in the ABE horizons confirm the results observed in other areas of ABEs (Lehmann et al., 2003, Cunha et al., 2016), revealing that soil OC plays a key role in providing cation exchange sites. The higher amount of CO in the ABEs gives the soil a higher CEC, corroborating with other investigations, which attribute the high specific surface area to the higher negative charge density per unit area, thus, high CEC values result from the higher charge density



by carbon unit (Liang et al., 2006) (Zech et al., 1990; Lehmann et al., 2003; Glaser, 2007). This is a remarkable feature of EBEs, where OC content has been reported about 1.5 times higher when compared to adjacent soils, not EBEs (Aquino et al., 2016).

Regarding, it is not only the amount of OC that is responsible for the high CEC, the quality of the OC really has a greater effect. Studies have shown that OM in ABE contains larger amounts of carboxylic groups and phenolic groups compared to surrounding soils, and therefore OC in ABE has a higher CEC than OC in natural soils (Zech et al., 1990; Liang et al., 2006). The results showed that the chemical quality of ABE is strongly dependent on soil organic carbon. Thus, it is important to map the spatial pattern of the OC and the chemical properties of the ABEs to establish specific zones of management and agricultural planning, since the knowledge of the spatial variability of soil attributes allows the more efficient and economic recommendation of fertilizers and correctives (Roger et al., 2014; Brito et al, 2018; Oliveira et al., 2018).

The coefficient of variation (CV), parameterized in the Warrick & Nielsen (1980) proposal, was low (CV ≤ 12%), medium (60% < CV < 12%), or high (CV ≥ 60%) depending on the attribute and depth (Table 1). Among the attributes, $K^+$, P, $Mg^{2+}$, and m at the three depths and $Al^{3+}$ at depths of 0.0–0.05 and 0.10–0.20 m presented a high variability, which may influence methods and amount of samples to determine them. The attribute $Al^{3+}$ at a depth of 0.10–0.20 m and H+Al, $Ca^{2+}$, SB, t, and T at all depths presented medium variability, contrasting with the values of pH, V, and OC, which were classified as having a low variability at the studied soil depths. However, although this measure of dispersion can predict and compare the variability of variables with different units, CV does not allow analyzing the spatial variability of soil attributes nor its spatial pattern, thus requiring the adoption of the geostatistical analysis (Isaaks & Srivastava, 1989).

Semivariogram parameter assessment indicated a spatial structure dependence (Table 2). Spherical model prevailed to the semivariogram adjustment for all attributes regardless of the depths, except for pH, which presented an exponential model at a depth of 0.0−0.05 m. According to Isaaks & Srivastava (1989), exponential models are better adjusted to erratic phenomena on a small scale, while spherical models describe properties with a high spatial continuity or less erratic phenomena at a short distance. Thus, these are the models that best fit soil properties (Gomes et al., 2017).

Exchangeable aluminum was the only attribute that did not present spatial dependence, a phenomenon called pure nugget effect (PNE), corroborating the results found by Oliveira et al. (2014) to phosphorus in ABEs under forest cultivation (Table 2 and Figure 2). According to Gomes et al. (2018), this indicates that $Al^{3+}$ is spatially independent and has a random distribution. In general, PNE is extremely important since it indicates the unexplained variability, which may be due to measurement errors or undetected microvariations considering the sampling distance used (Cambardella et al., 1994). This behavior indicates the need to increase, in future studies, the grid spacing between sampling points in order to detect the spatial dependence of $Al^{3+}$.



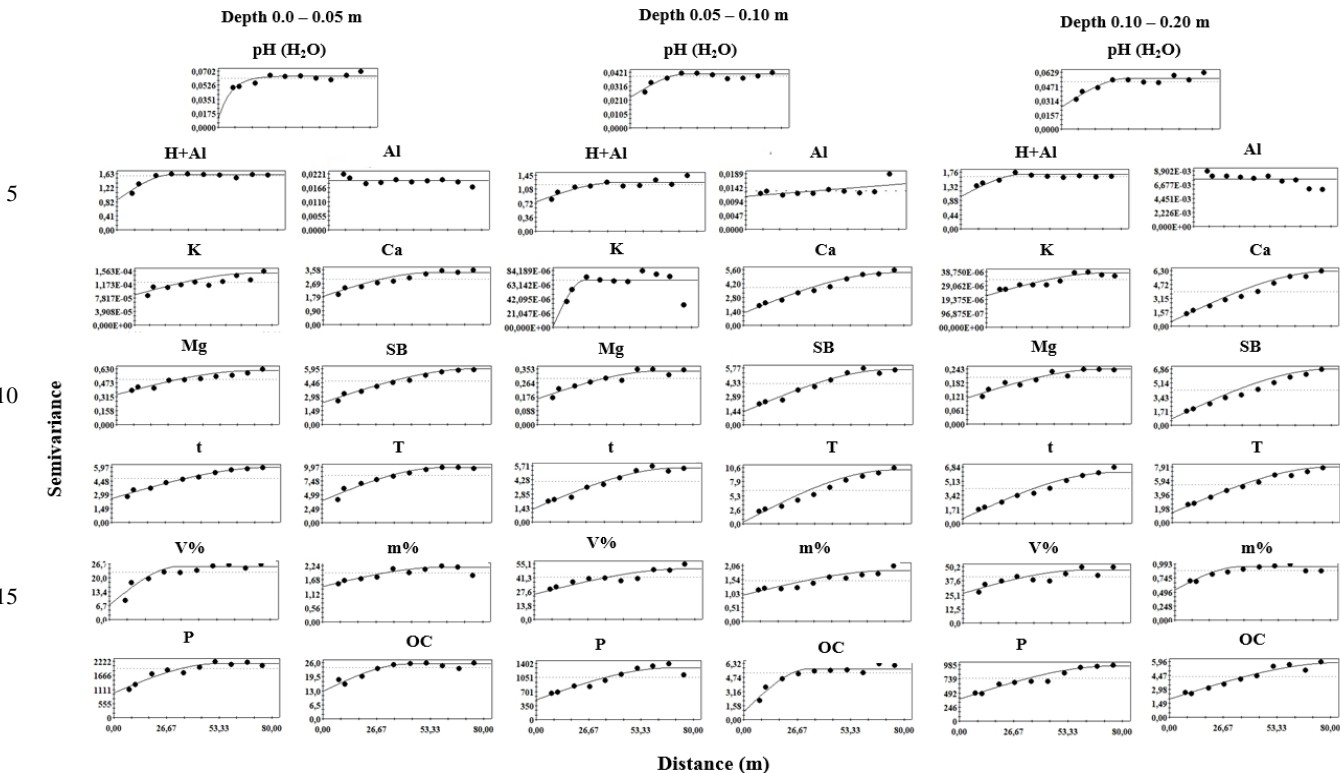

**Figure 2**: **Semivariograms of soil chemical attributes at depths of 0.0–0.05, 0.05–0.10, and 0.10–0.20 m in archeological black earth areas under pasture in Manicoré, AM, Brazil**





The attributes pH at a depth of 0.0–0.05 m, $K^+$, $Ca^{2+}$, SB, t, T, and OC at a depth of 0.05–0.10 m, and $Ca^{2+}$, SB, and T at a depth of 0.10–0.20 m presented a strong degree of spatial dependence (DSD), which is expressed by the ratio between the nugget effect ($C_0$) and range ($C_0+C_1$) (Cambardella et al., 1994). However, the attributes H+Al, $Al^{3+}$, $K^+$, $Ca^{2+}$, $Mg^{2+}$, SB, t, T,

5 V, m, P, and OC at a depth of 0.0–0.05 m, pH, H+Al, $Al^{3+}$, $Mg^{2+}$, V, m, and P at a depth 0.05–0.10 m, and pH, H+Al, $Al^{3+}$, $K^+$, $Mg^{2+}$, V, m, P, and OC presented a moderate DSD (Table 2). These results are similar to those found by Oliveira et al. (2014), who observed a moderate DSD for most of the studied attributes. According to Cambardella et al. (1994), the strong spatial dependence is an intrinsic soil property, a natural characteristic dependent on factors and processes of soil formation, while moderate and weak spatial dependences are more related to soil management, which can homogenize some soil

10 attributes.

The range values (Table 2 and Figure 2), which defines the maximum radius whose variable has spatial dependence (Trangmar et al., 1985), revealed a large amplitude of spatial variability, ranging from 20.40 to 79.40 m (0.0–0.05 m), 18.40 to 75.00 m (0.05–0.10 m), and 34.10 to 80.00 m (0.10–0.20 m). The range of 18.40 m indicated a greater spatial variability for $K^+$ (0.05–0.10 m) and smaller for T, which had a range of 80.00 m (0.10–0.20 m). In this sense, Guan et al. (2017) also found a greater

15 spatial discontinuity for $K^+$ every 17.0 m. In general, spatial variability was lower at a depth of 0.10–0.20 m, which indicates a natural behavior of soil, less influenced by pasture use.

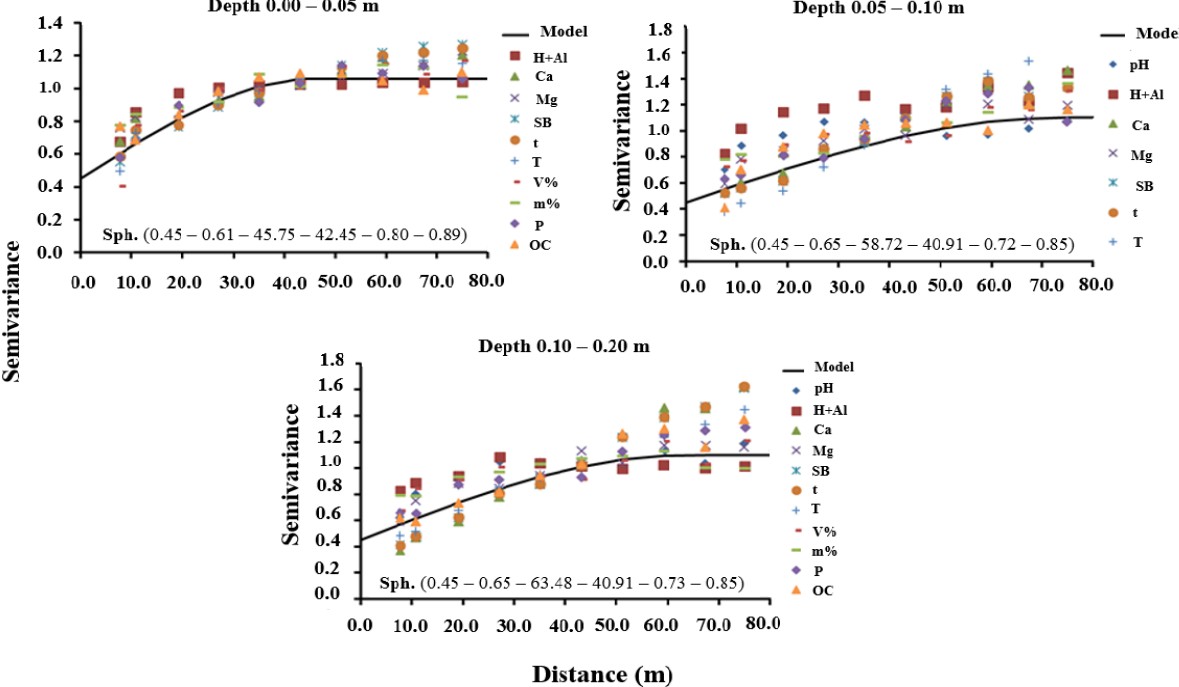

**Figure 3: Scaled semivariograms of chemical attributes at depths of 0.0–0.05, 0.05–0.10, and 0.10–0.20 m in archeological black earth areas under pasture in Manicoré, AM, Brazil. Values between brackets are, respectively, the**





**spherical model, C0 (nugget effect), C1 (sill), a (range, m), DSD (degree of spatial dependence), R2 (coefficient of determination), and CV (cross-validation).**

Based on the parameters of the scaled semivariograms (Figure 3), the spherical model best adjusted to all attributes, similarly to the individual semivariograms. Range values of 45.75 (0.0–0.05 m), 58.72 (0.0–0.10 m), and 63.48 m (0.10–0.20 m)

revealed that all attributes showed a greater spatial variability in the soil surface layer. Due to the greater spatial discontinuity at a depth of 0.0–0.05 m, the scaled semivariograms indicated a minimum sampling density of 2 points ha$^{-1}$, while depths of 0.05–0.10 and 0.10–0.20 m would require only 1 point ha$^{-1}$ to represent the spatial variability of soil chemical attributes.

The evidenced greater attribute heterogeneity at a depth of 0.0–0.05 m reinforces the anthropic influence associated with ABE formation. For some studies already developed in the study area, the greater spatial discontinuity in the surface layer is a

reflection of burning residues on the soil surface, especially when pasture is renewed (Gomes et al., 2017).

Maps of the spatial pattern of chemical attributes for depths of 0.0–0.05 m, 0.05–0.10 m, and 0.10–0.20 m are shown in Figure 4. The differentiated behavior of isolines in the respective depths indicates a management influence. At a depth of 0.0–0.05 m, less defined isolines for pH in relation to other depths not only show a greater spatial discontinuity but also reinforce their sensitivity to pasture management. The spatial pattern of K$^+$ was that most differed among depths, which is consistent with the

higher amplitude of range values (18.40 to 70 m). In general, the highest values of Ca$^{2+}$, P, SB, t, T, and m, regardless of the depth, and Mg$^{2+}$ at depths of 0.0–0.05 m and 0.05–0.10 m were observed on the left portion of the maps. This spatial pattern confirms the existence of different management zones in the study area.

These results highlight the importance of mapping soil attributes in order to establish a regional management of soil and pasture in ABE areas. In addition to the knowledge of the spatial variability of these attributes, sampling density allows the feasibility

of extending the study in large areas, for example to the Amazonas State, whose great territorial extension and geomorphological diversity would be a hindrance to the development of this study by traditional methods. The use of these techniques maximizes sampling efficiency, time, cost reduction, and manpower, enabling the development of detailed maps of the southern Amazon.





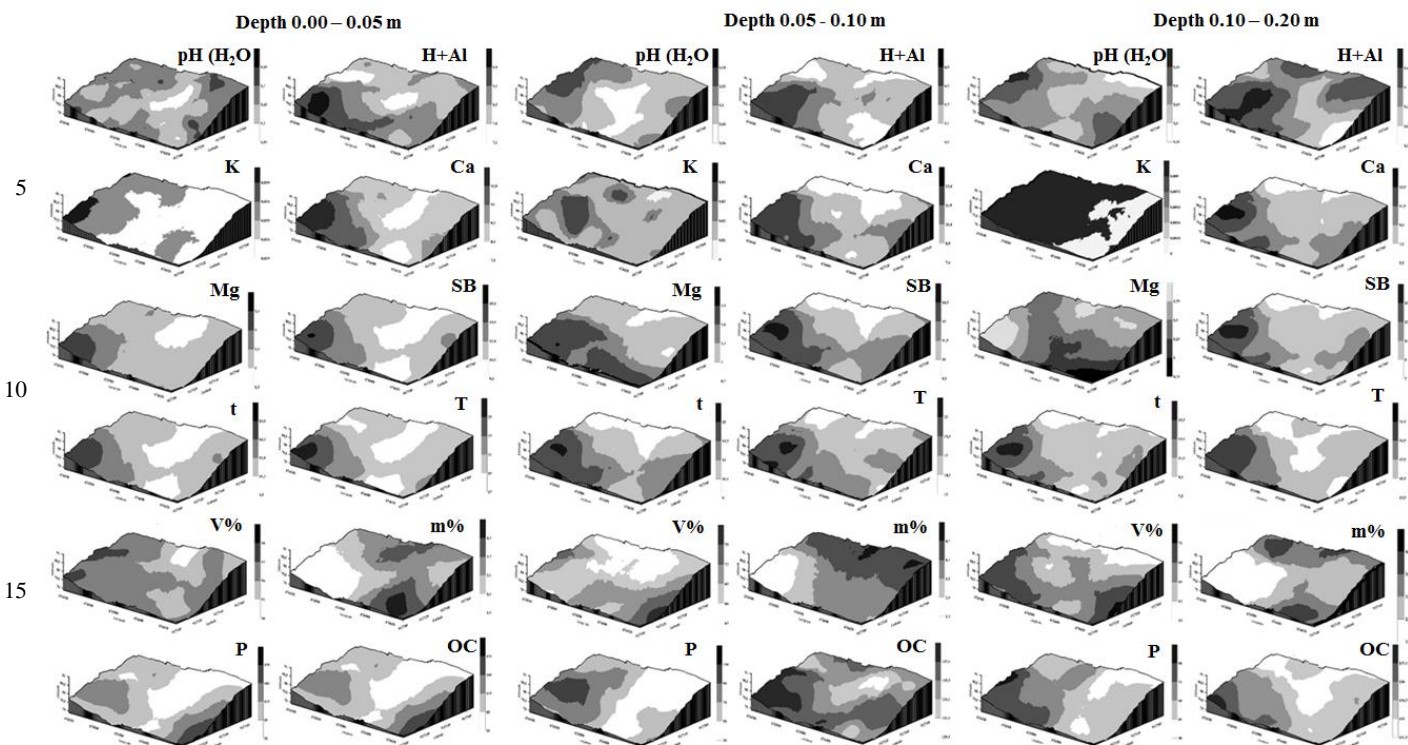

Figure 4: Kriging maps of chemical attributes at depths of 0.0–0.05, 0.05–0.10, and 0.10–0.20 m in archeological black earth areas under pasture in Manicoré, AM, Brazil.



## 4 Conclusion

The attributes present a spatial dependence varying from strong to moderate, with Al3+ being the only chemical attribute that does not present a spatial dependence structure at the assessed depths. Scaled semivariograms satisfactorily reproduce the spatial behavior of soil attributes following the same pattern of individual semivariograms, allowing their use to estimate the variability of soil attributes. Sampling density is higher at a depth of 0.0–0.05 m, requiring 2 and 1 point ha−1 for depths of 0.05–0.10 m and 0.10–0.20 m, respectively, to represent the spatial pattern of chemical attributes.

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





**Table 1: Descriptive statistics of soil chemical attributes at depths of 0.0–0.05, 0.05–0.10, and 0.10–0.20 m in archaeological black earth areas under pasture in Manicoré, AM, Brazil.**

| Descriptive statistics | pH (H₂O) | H+Al | Al³⁺ | K⁺ | Ca²⁺ | Mg²⁺ | SB | t | T | V% | m% | P mg dm⁻³ | OC g dm⁻³ |
|---|---|---|---|---|---|---|---|---|---|---|---|---|---|
| | (H2O) | -----------------------------cmol$_c$dm⁻³----------------------------- | | | | | | | | | | mg dm⁻³ | g dm⁻³ |
| **Depth** | | | | | | 0.0 – 0.05 m | | | | | | | |
| Maximum | 6.07 | 11.72 | 0.70 | 0.04 | 14.75 | 3.75 | 18.05 | 18.45 | 27.79 | 77.90 | 5.04 | 117.45 | 147.60 |
| Minimum | 4.88 | 4.62 | 0.20 | 0.01 | 5.50 | 1.00 | 7.76 | 7.96 | 12.87 | 41.66 | 1.27 | 20.77 | 116.41 |
| Mean | 5.47 | 8.51 | 0.39 | 0.02 | 9.28 | 2.29 | 11.61 | 11.98 | 20.14 | 57.64 | 2.73 | 75.17 | 137.07 |
| Median | 5.47 | 8.50 | 0.40 | 0.02 | 9.00 | 2.25 | 11.52 | 11.82 | 20.14 | 57.58 | 2.68 | 74.21 | 136.93 |
| ¹SD | 0.25 | 1.33 | 0.13 | 0.01 | 1.88 | 0.68 | 2.16 | 2.19 | 2.92 | 5.27 | 0.96 | 24.03 | 5.57 |
| Variance | 0.06 | 1.78 | 0.02 | 0.00 | 3.55 | 0.46 | 4.65 | 4.79 | 8.51 | 27.75 | 0.91 | 577.42 | 31.04 |
| ²CV% | 4.54 | 15.66 | 32.82 | 33.24 | 20.30 | 29.57 | 18.57 | 18.27 | 14.49 | 9.16 | 34.95 | 31.97 | 4.06 |
| Asymmetry | 0.06 | -0.08 | 0.69 | 0.20 | 0.55 | -0.02 | 0.38 | 0.35 | 0.22 | 0.35 | 0.57 | -0.10 | -0.72 |
| Curtose | -0.10 | -0.12 | -0.02 | -0.91 | 0.09 | -0.47 | -0.24 | -0.28 | -0.20 | 2.38 | -0.37 | -0.75 | 1.50 |
| ³d | 0.07* | 0.07* | 0.22* | 0.10* | 0.12* | 0.12* | 0.09* | 0.09* | 0.07* | 0.06* | 0.08* | 0.09* | 0.06* |
| **Depth** | | | | | | 0.05 – 0.10 m | | | | | | | |
| Maximum | 6.02 | 7.76 | 0.50 | 0.02 | 14.50 | 2.75 | 16.02 | 16.42 | 23.60 | 92.09 | 5.79 | 97.20 | 139.02 |
| Mínimum | 5.34 | 2.97 | 0.20 | 0.00 | 6.00 | 0.25 | 7.51 | 7.91 | 11.14 | 53.21 | 1.43 | 17.28 | 127.72 |
| Mean | 5.72 | 5.46 | 0.39 | 0.01 | 10.09 | 1.36 | 11.47 | 11.91 | 16.93 | 67.74 | 3.71 | 52.42 | 135.28 |
| Median | 5.76 | 5.61 | 0.40 | 0.01 | 9.75 | 1.50 | 11.26 | 11.76 | 16.75 | 67.66 | 3.67 | 51.67 | 135.56 |
| ¹SD | 0.18 | 1.15 | 0.08 | 0.00 | 1.92 | 0.52 | 2.05 | 2.03 | 2.55 | 6.48 | 1.07 | 21.90 | 2.28 |
| Variance | 0.03 | 1.32 | 0.01 | 0.00 | 3.68 | 0.27 | 4.19 | 4.13 | 6.48 | 42.00 | 1.15 | 479.82 | 5.22 |
| ²CV% | 3.07 | 21.04 | 19.64 | 40.73 | 19.03 | 38.37 | 17.83 | 17.05 | 15.03 | 9.57 | 28.95 | 41.79 | 1.69 |
| Asymmetry | -0.39 | 0.00 | -0.28 | 1.04 | 0.40 | 0.02 | 0.34 | 0.32 | 0.42 | 0.49 | 0.10 | 0.36 | -0.69 |
| Curtose | -0.77 | -0.72 | -0.32 | 0.95 | -0.37 | -0.17 | -0.35 | -0.37 | 0.19 | 1.72 | -0.84 | -0.90 | 0.42 |
| ³d | 0.10* | 0.07* | 0.27* | 0.13* | 0.09* | 0.12* | 0.09* | 0.08* | 0.08* | 0.09* | 0.08* | 0.13* | 0.09* |
| **Depth** | | | | | | 0.10 – 0.20 m | | | | | | | |
| Maximum | 6.36 | 7.43 | 0.50 | 0.01 | 15.25 | 3.00 | 17.76 | 18.16 | 24.69 | 77.48 | 5.26 | 90.24 | 141.76 |
| Mínimum | 5.35 | 3.14 | 0.20 | 0.00 | 4.75 | 0.25 | 6.51 | 6.81 | 12.04 | 49.64 | 1.36 | 17.21 | 129.72 |
| Mean | 5.73 | 5.44 | 0.32 | 0.00 | 9.41 | 1.47 | 10.89 | 11.22 | 16.62 | 65.48 | 2.92 | 50.43 | 133.91 |
| Median | 5.75 | 5.61 | 0.30 | 0.00 | 9.25 | 1.50 | 10.51 | 10.91 | 16.19 | 65.31 | 2.86 | 46.63 | 134.10 |
| ¹SD | 0.22 | 1.16 | 0.08 | 0.00 | 1.96 | 0.52 | 2.05 | 2.05 | 2.54 | 6.55 | 0.88 | 19.68 | 2.09 |
| Variance | 0.05 | 1.34 | 0.01 | 0.00 | 3.85 | 0.27 | 4.20 | 4.21 | 6.47 | 42.90 | 0.77 | 387.17 | 4.35 |
| ²CV% | 3.88 | 21.31 | 25.85 | 54.13 | 20.84 | 35.32 | 18.82 | 18.29 | 15.31 | 10.00 | 30.11 | 39.02 | 1.56 |
| Asymmetry | 0.45 | -0.23 | 0.16 | 0.79 | 0.47 | 0.09 | 0.69 | 0.7 | 0.57 | -0.15 | 0.33 | 0.43 | 0.51 |
| Curtose | -0.18 | -1.27 | -0.58 | -0.67 | 0.59 | 0.38 | 0.94 | 0.97 | 0.02 | -0.81 | -0.51 | -0.81 | 1.17 |
| ³d | 0.10* | 0.14* | 0.24* | 0.22* | 0.11* | 0.12* | 0.10* | 0.10* | 0.10* | 0.08* | 0.08* | 0.09* | 0.09* |

¹SD: standard deviation; ²CV: coefficient of variation; ³d: Kolmogorov-Smirnov normality test, *Significant at 5% probability.



**Table 2:** **Geostatistics of soil chemical attributes at depths of 0.0–0.05, 0.05–0.10, and 0.10–0.20 m in archeological black earth areas under pasture in Manicoré, AM, Brazil.**

| | pH | H+Al | Al$^{3+}$ | K$^+$ | Ca$^{2+}$ | Mg$^{2+}$ | SB | t | T | V% | m% | P | OC |
|---|---|---|---|---|---|---|---|---|---|---|---|---|---|
| | (H$_2$O) | ------------------------------cmol$_c$dm$^{-3}$-------------------------------- | | | | | | | | | | mg dm$^{-3}$ | g dm$^{-3}$ |
| | | | | | | 0.00 – 0.05 m | | | | | | | |
| Model | Exp. | Sph. | Lin. | Sph. | Sph. | Sph. | Sph. | Sph. | Sph. | Sph. | Sph. | Sph. | Sph. |
| C0 | 0.01 | 0.86 | - | 8.7x10$^-$ | 1.85 | 0.34 | 2.31 | 2.57 | 3.90 | 6.86 | 1.41 | 978.00 | 12.67 |
| C0 +C1 | 0.06 | 1.61 | - | 1.5x10$^-$ | 3.41 | 0.61 | 5.95 | 5.97 | 9.86 | 25.49 | 2.18 | 2.135.0 | 25.35 |
| a (m) | 20.40 | 28.00 | - | 70.00 | 55.00 | 70.00 | 76.00 | 79.40 | 59.50 | 35.00 | 60.00 | 53.10 | 42.40 |
| R$^2$ | 0.73 | 0.91 | - | 0.75 | 0.91 | 0.91 | 0.98 | 0.98 | 0.96 | 0.87 | 0.77 | 0.92 | 0.90 |
| DSD % | 16.67 | 53.42 | PNE | 58.00 | 54.25 | 55.74 | 38.82 | 43.05 | 39.55 | 26.91 | 64.68 | 45.81 | 49.98 |
| CV | 0.73 | 0.83 | - | 0.82 | 0.80 | 0.91 | 0.89 | 0.89 | 0.90 | 0.78 | 0.81 | 1.00 | 0.99 |
| | | | | | | 0.05 – 0.10 m | | | | | | | |
| Model | Sph. | Sph. | Lin. | Sph. | Sph. | Sph. | Sph. | Sph. | Sph. | Sph. | Sph. | Sph. | Sph. |
| C0 | 0.02 | 0.75 | - | 1.0x10$^-$ | 1.24 | 0.16 | 1.30 | 1.29 | 0.27 | 24.86 | 0.96 | 449.00 | 0.87 |
| C0 +C1 | 0.04 | 1.26 | - | 7.0x10$^-$ | 5.30 | 0.34 | 5.62 | 5.51 | 10.11 | 50.02 | 1.88 | 1.291.0 | 5.72 |
| a (m) | 30.00 | 40.00 | - | 18.40 | 75.00 | 63.20 | 73.90 | 72.90 | 75.00 | 75.00 | 75.00 | 68.90 | 33.10 |
| R$^2$ | 0.82 | 0.74 | - | 0.36 | 0.97 | 0.90 | 0.97 | 0.97 | 0.96 | 0.79 | 0.85 | 0.89 | 0.91 |
| DSD % | 50.00 | 59.52 | PNE | 1.43 | 23.40 | 47.06 | 23.13 | 23.41 | 2.67 | 49.70 | 51.06 | 34.78 | 15.21 |
| CV | 0.80 | 0.80 | - | 0.89 | 0.96 | 0.91 | 0.94 | 0.94 | 0.93 | 0.95 | 0.74 | 0.89 | 0.94 |
| | | | | | | 0.10 – 0.20 m | | | | | | | |
| Model | Sph. | Sph. | Lin. | Sph. | Sph. | Sph. | Sph. | Sph. | Sph. | Sph. | Sph. | Sph. | Sph. |
| C0 | 0.03 | 1.00 | - | 2.2x10$^-$ | 0.50 | 0.11 | 0.75 | 0.59 | 1.38 | 26.48 | 0.53 | 383.62 | 1.93 |
| C0 +C1 | 0.06 | 1.70 | - | 3.8x10$^-$ | 6.24 | 0.24 | 6.86 | 6.17 | 7.91 | 47.14 | 0.94 | 965.25 | 5.85 |
| a (m) | 35.00 | 32.00 | - | 70.00 | 80.00 | 69.20 | 80.00 | 75.00 | 80.00 | 60.00 | 34.10 | 75.00 | 79.10 |
| R$^2$ | 0.84 | 0.86 | - | 0.85 | 0.97 | 0.92 | 0.97 | 0.96 | 0.99 | 0.72 | 0.84 | 0.92 | 0.95 |
| DSD % | 50.00 | 58.82 | PNE | 57.89 | 8.01 | 45.83 | 10.93 | 9.56 | 17.45 | 56.17 | 56.38 | 39.74 | 32.99 |
| CV | 0.77 | 0.78 | - | 0.86 | 0.97 | 0.95 | 1.00 | 0.98 | 0.97 | 0.98 | 0.88 | 0.90 | 1.00 |

Sph: Spherical; Exp: Exponential; Lin: Linear; C0: nugget effect; C0+C1: sill; a: range (m); R2: coefficient of determination; DSD: degree of spatial dependence; and CV: cross-validation.