# Peer review of "Spatial variability and sampling density of chemical attributes in archaeological black earths under pasture in southern Amazonas, Brazil"

_SOIL, 2019_

## Referee Comment (RC1) · Anonymous Referee #1 · 8 Jul 2019

General comments

The authors sampled a plot in the Brazilian state of Amazonas and analysed the samples on chemical composition, to assess the spatial variability and thereby recommend a sampling density for everyday agricultural use. The topic is relevant, and the data seems abundant and interesting; however, from my – rather theoretical geostatistical - viewpoint, I see possibilities for improvement, especially with a clearer explanation and use of geostatistical concepts. I advise the authors to draw inspiration from a geostatistical textbook or for example from the summary paper by Oliver and Webster,

2014 (https://doi.org/10.1016/j.catena.2013.09.006), rather than rewording the some-what crudely presented theory as shown in very applied papers such as Gomes et al. (2017). For example, please make a clear distinction between 1) an experimental variogram, which follows directly from the data, but where perhaps a decision about bin width etc. has to be taken, 2) The variogram model, or mathematical variogram model, with a chosen covariance model and estimated parameters and 3) the prediction (or kriging). For example, on page 4, lines 21-24, it is not clear to me what you mean by "choosing" an experimental variogram model, and I suppose the $R2$ has to do with mathematical model fit, and cross-validation has to do with the kriging phase? Please make clear what is meant.

In the discussion or in the conclusion, I would expect some reflection on the main finding about the required sampling density. Do the authors expect the same spatial variability in similar soils with the same history, in similar soils with a different history, etc.? In how far can the findings be generalised?

Note that I am not qualified to say anything meaningful about soil chemistry and its agricultural consequences, so I will refrain from that topic.

General textual remark: In my pdf, several would-be subscripts in the text are actually normal, in the chemical formula as well as in the mathematical variables.

Specific comments and textual remarks

Page 1, line 20 and also page 3, line 2: Perhaps use "required sampling density" rather than just "sampling density".

Page 2, line 33: ", Mapping" should not have a capital

Page 3, Figure 1: Please explain why DEM is shown. As I understood it, it is not used as explanatory variable.

Page 3, line 17: To maintain which quality? Of the crop, or the soil?

Page 3, line 19: An GPS accuracy up to 15 m, when the sample locations are supposed to be 8 m apart, seems quite problematic. Perhaps elaborate in the discussion which influence this might have on the results and conclusions, or explain better what it actually means.

Page 4, line 15/16: The line "Based on the assumption of stationarity of the intrinsic hypothesis.." is a bit vague, which hypothesis is meant?

Page 5, line 3-8, and also page 9, figure 3: I am sorry, but I don't understand what is meant here. With scaling, the nugget-sill ratio will not change. Please elaborate, also because the referred paper [Vieira (1997)] seems to be in Portuguese.

Page 5, equation (3): As this leads to the central conclusion of this paper, please explain where this formula comes from, and perhaps also when it is applicable and which are the underlying assumptions.

Page 5, line 21-22: Which "Values close to.." are meant?

Page 7, line 21-23: I am not sure about the assumed short distance differences between a spherical and an exponential variogram model. Perhaps the authors confused "Gaussian" with "spherical", based on the comparison in Isaaks & Srivastava (1989), page 303?

Page 7, line 23: Firstly: what is meant by "these" models? Exponential, or spherical? Secondly, it is quite a strong statement that a certain model "best fits soil properties". How is this information extracted from the paper of Gomers at al. (2017) – as I could not find it - , and in which context is this statement valid? Also note that a spherical variogram model has a range which is equal to the range parameter, but an exponential variogram model has a so-called "effective range" being 3x the range parameter; this should be taken into account when comparing

Page 7, line 26-30: The words ".. has a random distribution" might give confusion, as "random" does not mean "not correlated". Also the sentence "In general, .... used"

can be phrased more to the point. A pure nugget variogram indicates that there is no spatial structure found with the given sampling scheme. And why would one expect to find a spatial dependence in increased grid spacing? This might happen, but perhaps rather with smaller grid spacing, or perhaps not at all.

Page 11, figure 4: Perhaps, also mention the kriging error uncertainty (the kriging variance), as that is a substantial part of geostatistics.

---

## Author Comment (AC1) · 6 Aug 2019

All comments were rigorously analyzed and followed to ensure increased quality of work. This way I am submitting the revised article. Equation 3: It is widely used to define the sample density to be used, where the stepped semivariogram is interpolated and from this equation the number of samples per area is defined, this based on the second sentence of the conclusion paragraph, in relation to the sample density.

Please also note the supplement to this comment:

https://www.soil-discuss.net/soil-2019-26/soil-2019-26-AC1-supplement.pdf

**Supplement:**

https://doi.org/10.5194/soil-2019-26

This is just a preview and not the published preprint.
Ṡ Author(s) 2019. CC-BY 4.0 License.

[revised manuscript text omitted]

https://doi.org/10.5194/soil-2019-26
This is just a preview and not the published preprint.

ⓢ Author(s) 2019. CC-BY 4.0 License.

[Figure]

[Figure]

[Figure]

**Figure 4: Kriging maps of chemical attributes at depths of 0.0–0.05, 0.05–0.10, and 0.10–0.20 m in archeological black earth areas under pasture in Manicoré, AM, Brazil.**

https://doi.org/10.5194/soil-2019-26
This is just a preview and not the published preprint.
§ Author(s) 2019. CC-BY 4.0 License.

[Figure]

[Figure]

**4 Conclusion**

The attributes present a spatial dependence varying from strong to moderate, with Al3+ being the only chemical attribute that does not present a spatial dependence structure at the assessed depths. Scaled semivariograms satisfactorily reproduce the spatial behavior of soil attributes following the same pattern of individual semivariograms, allowing their use to estimate the variability of soil attributes. Sampling density is higher at a depth of 0.0–0.05 m, requiring 2 and 1 point ha−1 for depths of 0.05–0.10 m and 0.10–0.20 m, respectively, to represent the spatial pattern of chemical attributes. Due to the greater anthropic action on the surface, spatial variability tends to decrease in depth.

$\mathcal{S}$ Author(s) 2019. CC-BY 4.0 License.

[revised manuscript text omitted]

---

## Referee Comment (RC2) · Anonymous Referee #1 · 17 Oct 2019

Dear authors, thanks for your response, although it could have been more informative. Additional to that, I used dedicated software to compare the previous manuscript to the latest version, so see in what way you followed my suggestions. I found 4 small textual changes, 1 changed and 3 added references, and one informative sentence added. I am sorry to say that in my opinion, this manuscript in its current state is still far from ready for publication. Most of my earlier comments are not addressed: not in the manuscript, not in your response. I suggest to withdraw or reject this manuscript, until the geostatistical part has been seriously improved.

---

## Referee Comment (RC3) · Anonymous Referee #2 · 2 Feb 2020

The paper presents a spatial analysis based on variograms for several soil properties in black earths in Amazonas. Although I agree with the authors that this is an interesting soil type, with relevance for agriculture and climate change, I found the paper very descriptive and only of local relevance. The spatial variability for a specific micro test site on pasture was assessed, but it is unclear how representative this site is. Furthermore, the spatial variability seems to be controlled topography (fig 4), suggesting but the factors controlling spatial patterns of soil properties are not addressed here. As such, the paper is not ready for publication because of (i) the lack of specific scientific questions

and hypothesis to be tested, and (ii) the local focus, without any consideration for the broader significance.

---

## Author Comment (AC2) · 17 Feb 2020

Since the first version, changes were made to the manuscript, in order to improve and improve the material, as requested by the discussion. Geostatistics was seen again and information was added without prolonging it in a way that makes reading the text tiring. In this way I present the final version of the work with the changes.

---

## Author Comment (AC3) · 17 Feb 2020

he manuscript underwent changes in its textual body and handling of the geostatistical tool, so that it could fit the profile of the magazine, as well as improving the quality of its text, so I believe that after this it would be in agreement to be able to be incorporated into magazine, since it is a study on soils of the Amazonian environment, which worldwide is a well-known biome and of natural and environmental importance, as well as the study is carried out on soils formed from the anthropization of indigenous peoples traces of the history of formation of the region's soils. So I think it would attract

attention and at the same time bring important information about this biome.

Please also note the supplement to this comment:
https://www.soil-discuss.net/soil-2019-26/soil-2019-26-AC3-supplement.pdf

**Supplement:**

https://doi.org/10.5194/soil-2019-26
This is just a preview and not the publishedpreprint.
Ⓢ Author(s) 2019. CC-BY 4.0 License.

[revised manuscript text omitted]

Thus, when crossing information from the DSD with the range, it is possible to state that the greater the degree of spatial dependence and the lower the range, the greater the variation of this attribute in the established space. Because if the reach is low it means the little uniformity of the attribute in the place, as well as if the degree of dependence is weak, as it is

https://doi.org/10.5194/soil-2019-26
This is just a preview and not the publishedpreprint.
§ Author(s) 2019. CC-BY 4.0 License.
5   indicative of greater dispersion, in this case it is recommended to decrease the sample grid and thus be able to have a better

spatial evaluation of the attribute (Dalchiavon et al., 2017).

[revised manuscript text omitted]

https://doi.org/10.5194/soil-2019-26
This is just a preview and not the publishedpreprint.
Ⓢ Author(s) 2019. CC-BY 4.0 License.

[Figure]

[Figure]

5   https://doi.org/10.1016/j.geoderma.2015.07.010, 2016.

Barbosa, R. D., de Lima, A. F. L., Simões, E. L., de Brito Filho, E. G., Campos, M. C. C., da Cunha, J. M., Souza, F. G. (2019). Spatial variation of chemical attributes in archaeological dark earth under cocoa cultivation in Western Amazon. Bioscience Journal, 35, 27-41, DOI: https://doi.org/10.14393/BJ-v35n1a2019-37137, 2019.

Brito, W.B.M., Campos, M.C.C., Mantovanelli, B.C., Cunha, J.M., Franciscon, U., and Soares, M.D.R. Spatial variability of

10   soil physical properties in Archeological Dark Earths under different uses in southern Amazon. Soil and Tillage Research, 182, 103-111, DOI: 10.1016/j.still.2018.05.008, 2018.

Brito Filho E. G., Mantovanelli, B. C., Brito, W. B. M., Silva, J. F., Campos, M. C. C., Cunha, J. M. Variabilidade espacial da textura do solo em área de terra preta arqueológica sob diferentes usos na Região Sul do Amazonas. Scientia Agraria Paranaensis, v. 17, 139-143, 2018.

15   Burrough, P.A. and McDonnel, R.A. Principles of geographical information systems. Oxford University Press, 2000.

Cambardella, C.A., Moorman, T.B., Novak, J., Parkin, T.B., Karlen, D.L., Turco, R.F. and Konopka, A.E. Field-scale variability of soil properties in Central Iowa Soils. Soil Science Society of America Journal, 58, 1501–1511, DOI: 10.2136/sssaj1994.03615995005800050033x, 1994.

Campos, M.C.C., Oliveira, I.A., Santos, L.A.C., Aquino, R.E. & Soares, M.D.R. Variabilidade espacial da resistência do solo

20   à penetração e umidade em áreas cultivadas com mandioca na região de Humaitá, AM. Revista Agro@mbiente On-line, 6, 09–16, http://dx.doi.org/10.18227/1982-8470ragro.v6i1.689, 2012.

Campos, M.C.C., Ribeiro, M.R., Souza Júnior, V.S., Ribeiro Filho, M.R., Souza, R.V.C.C. and Almeida, M.C. Caracterização e classificação de Terras Pretas Arqueológicas na região do Médio Rio Madeira. Bragantia, 70, 18–27, http://dx.doi.org/10.1590/S0006-87052011000300016, 2011.

25   Dalchiavon, F. C., Rodrigues, A. R., de Lima, E. S., Lovera, L. H., & Montanari, R. Variabilidade espacial de atributos químicos do solo cultivado com soja sob plantio direto. Revista de Ciências Agroveterinárias, 16, 144-154, 2017.

Embrapa – Empresa Brasileira de Pesquisa Agropecuária. Centro Nacional de Pesquisa de Solos. Manual de métodos de análise de solo. Embrapa, 2011.

Embrapa – Empresa Brasileira de Pesquisa Agropecuária. Centro Nacional de Pesquisa de Solos. Sistema brasileiro de

30   classificação de solo. Embrapa, 2012.

Falcão, N.P.S. and Borges, L.F. Efeito da fertilidade de terra preta de índio da Amazônia Central no estado nutricional e na produtividade do mamão hawaí (Caricapapaya L.). Acta Amazônica, 36, 401–406, http://dx.doi.org/10.1590/S0044-59672006000400001, 2006.

Glaser, B. and Birk, J.J. State of the scientific knowledge on properties and genesis of Anthropogenic Dark Earths in Central

35   Amazonia (Terra Preta de Índio). Geochimica et Cosmochimica Acta, 82, 39–51, https://doi.org/10.1016/j.gca.2010.11.029, 2012.

Glaser, B. Prehistorically modified soils of central Amazonia: a model for sustainable agriculture in the twenty-first century. Philosophical Transactions of the Royal Society B, 362, 187 – 196, https://dx.doi.org/10.1098%2Frstb.2006.1978, 2007.

Glaser, B., Haumaier, L., Guggenberger, G., and Zech, W. The Terra Preta phenomenon: a model for sustainable agriculture

40   in the humid tropics. Naturwissenschaften. 88: 37-41, DOI: 10.1007/s001140000193, 2001.

Gomes, R.P., Campos, M.C.C., Brito, W.B.M., Cunha, J.M., Muniz, A.W., Silva, L.S., Souza, E. D., Oliveira, I.A., and Freitas,

https://doi.org/10.5194/soil-2019-26
This is just a preview and not the publishedpreprint.
Ṡ Author(s) 2019. CC-BY 4.0 License.

[revised manuscript text omitted]